# The Role of RNA-Binding Proteins in Vertebrate Neural Crest and Craniofacial Development

**DOI:** 10.3390/jdb9030034

**Published:** 2021-08-27

**Authors:** Thomas E. Forman, Brenna J. C. Dennison, Katherine A. Fantauzzo

**Affiliations:** Department of Craniofacial Biology, School of Dental Medicine, University of Colorado Anschutz Medical Campus, Aurora, CO 80045, USA; thomas.forman@cuanschutz.edu (T.E.F.); brenna.dennison@cuanschutz.edu (B.J.C.D.)

**Keywords:** RNA-binding protein, neural crest, craniofacial development

## Abstract

Cranial neural crest (NC) cells delaminate from the neural folds in the forebrain to the hindbrain during mammalian embryogenesis and migrate into the frontonasal prominence and pharyngeal arches. These cells generate the bone and cartilage of the frontonasal skeleton, among other diverse derivatives. RNA-binding proteins (RBPs) have emerged as critical regulators of NC and craniofacial development in mammals. Conventional RBPs bind to specific sequence and/or structural motifs in a target RNA via one or more RNA-binding domains to regulate multiple aspects of RNA metabolism and ultimately affect gene expression. In this review, we discuss the roles of RBPs other than core spliceosome components during human and mouse NC and craniofacial development. Where applicable, we review data on these same RBPs from additional vertebrate species, including chicken, *Xenopus* and zebrafish models. Knockdown or ablation of several RBPs discussed here results in altered expression of transcripts encoding components of developmental signaling pathways, as well as reduced cell proliferation and/or increased cell death, indicating that these are common mechanisms contributing to the observed phenotypes. The study of these proteins offers a relatively untapped opportunity to provide significant insight into the mechanisms underlying gene expression regulation during craniofacial morphogenesis.

## 1. Introduction

During vertebrate embryogenesis, neural crest (NC) cells are induced at the neural plate border, undergo an epithelial to mesenchymal transition to delaminate from the neural folds or neural tube, migrate in stereotypical streams throughout the embryo and eventually differentiate into a wide array of derivatives. The NC can be subdivided into four axial populations: cranial, cardiac, vagal and trunk. Cranial NC cells originate from the forebrain to the hindbrain and migrate into the frontonasal prominence and pharyngeal arches. These cells generate the bone and cartilage of the frontonasal skeleton, cartilages of the jaw, middle ear, hyoid and thyroid, tendons, smooth muscle, connective tissue, melanocytes and cranial sensory ganglia of the peripheral nervous system. Cranial NC cells regulate growth and patterning of the brain, and additionally contribute to the eyes, teeth, thyroid gland, parathyroid glands and thymus [1].

RNA-binding proteins (RBPs) have emerged as critical regulators of NC and craniofacial development. Conventional RBPs bind to specific sequence and/or structural motifs in a target RNA via one or more RNA-binding domains (RBDs) to regulate multiple aspects of RNA metabolism, including transcription, splicing, modification, trafficking, translation and decay, and ultimately affect gene expression. A growing body of evidence has also identified unconventional RBPs that do not contain discernible RBDs [2,3].

In this review, we discuss the roles of RBPs other than core spliceosome components during human and/or mouse NC and craniofacial development. These include a regulator of transcription, CNBP; proteins involved in splicing such as EIF4A3, ESRP1/2, FAM50A, hnRNP H2, hnRNP K, hnRNP U, Hu proteins, PTB, RBFOX2, RBM10, RBMX and SRSF3; a regulator of nuclear export, EIF4A3; regulators of trafficking such as EIF4A3 and IMP-1; proteins involved in translation such as CNBP, DDX3X, EIF4A3, FMRP, PACT and IMP-1; and a regulator of decay, EIF4A3 (Figure 1). These proteins contain one or more RBDs, including: RNA recognition motif (ESRP1/2, hnRNP H2, Hu proteins, PTB, RBFOX2, RBM10, RBMX, SRSF3 and IMP-1), zinc-finger domain (CNBP, hnRNP H2 and RBM10), K-homology domain (hnRNP K, IMP-1 and FMRP), DEAD motif (EIF4A3 and DDX3X), ROKNT domain (hnRNP K), SAP domain (hnRNP U), Fox-1_C domain (RBFOX2), G-patch domain (RBM10), RBM1CTR domain (RBMX), Agenet domain (FMRP), FXR1P_C domain (FMRP) and double-stranded RNA-binding motif (PACT) [4]. For a review of mutations associated with craniofacial spliceosomopathies, please see [5]. Where applicable, we review data on these same RBPs from additional vertebrate species, including chicken, *Xenopus* and zebrafish models, which provide further insight into the mechanisms by which these proteins regulate gene expression and cell activity. 

## 2. RBPs Involved in Transcription 

### CNBP

Myotonic dystrophy 2 (DM2; also known as proximal myotonic myopathy; OMIM 602668) is caused by a heterozygous expansion of 75 or more CCTG repeats in intron 1 of *CNBP* [6]. DM2 is characterized by muscle pain and stiffness, myotonia, progressive extremity and truncal weakness, cataracts, diabetes, cardiac arrhythmias and male hypogonadism, and may also include cognitive dysfunction, hearing loss, tremor and hypersomnia [7,8] (Table 1). Of note, in addition to its role in transcription, CNBP is also involved in translation.

In mice, *Cnbp* transcripts are expressed in the epiblast and visceral endoderm at embryonic day (E) 5.5. Approximately two days later, *Cnbp* transcripts and protein localize to all three germ layers in the anterior region of the embryo, including the anterior definitive endoderm underlying the future forebrain and the overlying anterior neuroectoderm. By E8.5, *Cnpb* expression is restricted to the headfold region. As early as E9.0, *Cnbp* is expressed in the forebrain, facial prominences, limb buds and somites. Expression in the craniofacial region is specifically localized to the frontonasal, maxillary and mandibular prominences and the roof of the stomodeum [9,10]. Mice that are homozygous for an allele harboring a viral insertion in intron 1 of *Cnbp* that abolishes transcript expression from the locus die around E10.5. These embryos display growth retardation as early as E7.5 with absence of the anterior mesendoderm and anterior neuroectoderm, and resulting truncations in the anterior neural folds at E8.5 and the forebrain at E9.5. These defects are associated with reduced proliferation in the anterior region of homozygous mutant embryos at E7.5–E8.5 and decreased expression of *Myc* in the anterior neuroectoderm and headfold region at E7.25 and E8.5, respectively. Further, approximately 40 percent of heterozygous neonatal pups exhibit growth retardation and craniofacial defects, including a small mandible and absence of the eyes [9]. Consistent with the phenotypes observed in human patients with DM2, mice that are heterozygous or homozygous for null alleles of *Cnbp* additionally exhibit postnatal muscle atrophy [11,12].

Similar to the expression patterns observed in mouse embryos, *CNBP* transcripts in the chick initially localize to the epiblast and hypoblast at stages XIII-XIV. At stage 4, *CNBP* is expressed in the neuroectoderm of the future forebrain. Expression is restricted to the anterior region of the embryo at stage 5, with strong expression in the anterior neural plate by stage 8. *CNBP* transcripts are detected in the telencephalon, midbrain, facial prominences, heart, wing bud and tail as early as stage 19. Akin to the findings detailed above for *Cnbp* homozygous mutant mouse embryos, injection of siRNA targeting *CNBP* into the chick forebrain at stage 9 leads to forebrain truncation [13].

In *Xenopus laevis*, *cnbp* is expressed throughout the animal pole and in the marginal zone containing the future mesoderm at early gastrula stages 10.25–10.5. During the late neural and tailbud stages 15–27, *cnbp* transcripts localize along the anterior to posterior dorsal axis, with specific expression in the neural tube, epidermis and somitic and lateral plate mesoderm [14]. Overexpression of *cnbp* mRNA leads to an expansion of the expression domain of the NC markers *foxd3* and *cmyc*. Alternatively, overexpression of transcripts encoding dominant-negative, N-terminal truncations of Cnbp results in decreased expression of *foxd3* and *cmyc* [15].

Similar to the findings in mice and chick, zebrafish *cnbpa* is expressed in the anterior region of the embryo, with increased expression at the midbrain-hindbrain border and the retina at 1 day post fertilization (dpf). At 2.5–3 dpf, *cnbpa* transcripts additionally localize to the craniofacial structures, pectoral fins and liver [16]. The *cis* elements that control zebrafish *cnbpa* expression during development are located in intron 1, in a region harboring conserved binding sites for NC transcription factors Pax6 and Foxd3 [17]. Knockdown of *cnbpa* at the one-cell stage with a translation-blocking morpholino results in growth retardation, abnormal morphology of the forebrain, midbrain and midbrain-hindbrain border, hydrocephaly, defects in craniofacial development, small eyes, fewer pigmented cells, oedema and tail curvature. The craniofacial defects include reduced first and second pharyngeal arch and neurocranial cartilage elements, disruption of pharyngeal arch 3–7 cartilages, a dysmorphic anterior ethmoid plate with large, round chondrocytes, decreased chondrocyte number in the trabeculae, malformation of the ceratohyal, reduction of the hyosymplectic and palatoquadrate and failed extension of Meckel’s cartilage at 4 dpf [16,18]. Overexpression of an N-terminal truncation of Cnbpa phenocopies knockdown with the translation-blocking morpholino [18]. These defects are associated with decreased cell proliferation and increased cell death in anterior embryonic regions at 1 dpf, particularly in the brain [16]. Further, decreased expression of the NC marker *msxb* is detected at 12 hours post fertilization (hpf) in morphant embryos, as well as decreased expression of the NC markers *foxd3, tfap2a, crestin, sox9b* and *dlx2a* in anterior embryonic regions at 1 dpf [16,18]. Moreover, expression of the cartilage marker *col2a1* is reduced in the parachordals and ceratohyal of morphants at 2 dpf and in the ethmoid plate and trabeculae at 3 dpf [16]. Knockdown of *cnbpa* at the one-cell stage with a morpholino that modifies *cnbpa* pre-mRNA splicing generates a more severe phenotype than that observed with the translation-blocking morpholino discussed above [18]. Interestingly, morpholino-mediated knockdown of the gene underlying the craniofacial disorder Treacher Collins Syndrome, *tcof1*, results in decreased levels of Cnbpa protein and increased levels of reactive oxygen species at 1 dpf. The craniofacial defects in these embryos at 4 dpf can be rescued upon transgenic overexpression of *cnbpa*, particularly, the lengths of the head, palatoquadrate-hyosymplectic cartilages and ceratohyal cartilages, and the distance between the ceratohyal cartilages joint and lateral fins. This rescue appears to stem from a protective role of Cnbpa in preventing the upregulation of oxidative stress-responsive genes [19].

## 3. RBPs Involved in Splicing 

### 3.1. EIF4A3

Robin sequence with cleft mandible and limb anomalies (also known as Richieri-Costa-Pereira syndrome; OMIM 268305) is caused by a homozygous 16-repeat expansion or compound heterozygous 16-repeat and 15-repeat expansions of an 18- or 20-nucleotide motif in the 5′ untranslated region (UTR) of *EIF4A3* [20]. The syndrome is characterized by craniofacial and limb defects, laryngeal abnormalities and, commonly, learning and language disabilities. The craniofacial defects include microstomia, cleft palate, absent lower central incisors, micrognathia, abnormal fusion of the mandible and ear anomalies [21,22] (Table 1). Induced pluripotent stem cell-derived NC cells (iNCCs) generated from patients with Robin sequence with cleft mandible and limb anomalies have approximately 70% and 50% reductions in the expression of *EIF4A3* mRNA and protein, respectively, and exhibit significant defects in migration. Mesenchymal stem-like cells generated from patient iNCCs have an approximately 15% reduction in *EIF4A3* expression and experience impaired chondrogenic differentiation and premature osteogenic differentiation compared to control cells [23]. Of note, in addition to its role in splicing, EIF4A3 is also involved in export, trafficking, translation and decay.

Similar to patients with Robin sequence with cleft mandible and limb anomalies, animal models with reduced *Eif4a3* expression display lower jaw defects. In mice, heterozygous, ubiquitous ablation of *Eif4a3* using a *Cmv-Cre* driver results in a reduction in body size, facial malformations, delayed eye and limb development and neural tube defects at E11.5. A subset of these embryos exhibits holoprosencephaly and a mandible lacking two distinct mandibular arches, consistent with expression of EIF4A3 in the developing mandible. Heterozygous, conditional ablation of *Eif4a3* in the NC lineage using the *Wnt1-Cre* driver leads to micrognathia at E11.5, a near complete absence of the skull, anterior facial skeletal elements and ears at E14.5, and a severely hypoplastic mandible, exencephaly, reduced eyelid closure and loss of the outer ear structures at E18.5 [23]. 

Similarly, morpholino-mediated knockdown of *eif4a3* in zebrafish results in craniofacial defects, including underdevelopment of craniofacial cartilage and bone elements and clefting of the lower jaw, as well as a reduction in the size of the eyes, underdevelopment of the third-sixth branchial arches and brain defects at 1 dpf. These changes are accompanied by increased apoptosis in morpholino-injected embryos, particularly in the anterior-most regions of the embryo, and decreased expression of transcripts encoding the NC markers Sox9b, Foxd3, Sox10 and Tbx2 [20].

### 3.2. ESRP1/2

Deafness, autosomal recessive 109 (DFNB109; OMIM 618013), characterized by bilateral sensorineural hearing loss and vestibular dysplasia, is associated with compound heterozygosity for a 19-bp deletion and a missense mutation in *ESRP1* (Table 1). Induced pluripotent stem cells generated from one of these patients exhibit significant changes in RNA splicing for a subset of ESRP1-dependent transcripts [24]. Alternatively, ESRP2 protein is expressed in the secondary palatal epithelium at 67–72 days of gestation in humans, with increased expression in the periderm. Relatedly, missense and nonsense mutations in *ESRP2* are associated with non-syndromic cleft lip with or without cleft palate (Table 1). Expression of the nonsense mutation in human embryonic kidney (HEK) 293T cells results in a partial disruption of epithelial RNA splicing [25].

In mice, *Esrp1* is expressed in the embryonic ectoderm as early as E6.5 and in the headfolds, open neural folds, definitive endoderm and foregut at E7.5. By E8.5, expression expands into the first pharyngeal arch and first pharyngeal pouch. *Esrp1* transcripts are additionally detected in the pharyngeal cleft and otic vesicle at E9.5, among other sites. By E10.5, *Esrp1* expression in the head region expands to Rathke’s pouch, the olfactory pit and the nasopharyngeal groove [26]. At E11.5, both *Esrp1* and *Esrp2* transcripts are expressed in the surface ectoderm of the medial and lateral nasal processes as well as the maxillary processes, with increased expression in the periderm [27]. By E12.5, *Esrp1* transcripts are additionally present in the lens, vibrissae and pinnae, among other sites [26]. From E13.5-E15.0, *Esrp1* and *Esrp2* are co-expressed in the epithelium lining the secondary palatal shelves [27]. Postnatally, *Esrp1* and *Esrp2* expression is restricted to epithelial cells across a wide range of tissues and organs [28,29]. Global ablation of *Esrp1* leads to fully penetrant, bilateral cleft lip and clefting of the primary and secondary palates, as well as defects in inner ear morphology, auditory hair cell differentiation and cell fate specification along the lateral wall of the cochlear epithelium [24,29]. Consistently, *Esrp1*-null embryos have reduced proliferation of both epithelial and mesenchymal cells in the medial and lateral nasal processes at E10.5, hypoplastic facial processes near mid-gestation, and persistent E-cadherin expression upon contact of the medial and lateral nasal processes at E11.5. Similarly, proliferation is reduced in the secondary palatal shelves of *Esrp1*-null embryos at E16.5 and palatal organ cultures established from these E13.5 embryos have persistent E-cadherin expression and medial edge epithelium presence upon contact of the shelves. These defects are accompanied by a switch in transcripts encoding ectodermal FGFR2IIIb to mesenchymal FGFR2IIIc in the facial process ectoderm at E12.0, rendering these cells incapable of responding to mesenchymal FGF10, in addition to decreased expression of transcripts encoding WNT ligands and SHH. Corresponding reductions in the expression of transcripts encoding canonical WNT targets and GLI transcription factors are detected in the facial process mesenchyme, as well as decreased WNT activity in the medial nasal and maxillary processes. Decreased expression of additional genes that have been implicated in facial clefting, such as *Bmp7*, *Tgfb2* and *Tgfb3*, were also found in the *Esrp1*-null facial process mesenchyme, indicating that ESRP1 serves to regulate signaling crosstalk between the facial process epithelium and mesenchyme [30]. Similarly, aberrant splicing of *Fgfr2* is detected in the cochlear epithelium of *Esrp1*-null embryos at 16.5 [24]. The same cleft lip and palate phenotypes observed in *Esrp1*-null embryos are recapitulated upon conditional ablation of *Esrp1* in the surface ectoderm using the *Crect* driver, such that these E18.5 embryos exhibit hypoplasia of the premaxilla and palatine bones and dysmorphic palatal process of the maxilla bones [30]. Global, simultaneous ablation of both *Esrp1* and *Esrp2* additionally results in absent lungs and salivary glands and reduced kidney size at E15.5, and reduced body length, rostral shortening, absent premaxillary bone and mandibular coronoid process, hypoplasia of the palatal bone and mandible, dorsal skin epidermal hypoplasia, reduced hair follicle number, delayed hair follicle maturation and forelimb malformations at E18.5. Again, a switch in transcripts encoding ectodermal FGFR2IIIb to mesenchymal FGRF2IIIc is detected in E18.5 epidermis obtained from double knockout embryos [29].

In zebrafish, *esrp1* and *esrp2* transcripts co-localize in the olfactory epithelium, otic vesicle, pharynx, epidermis and notochord from 1–5 dpf, with *esrp2* transcripts additionally expressed in the pronephros, hatching gland, liver and heart [31]. Both transcripts localize to the developing stomodeum and surface and oral epithelium at 2–3 dpf [27]. Combined loss-of-function of both *esrp1* and *esrp2* results in dysmorphology of the olfactory epithelium, clefting of the ethmoid plate, inner ear defects, abnormal basibranchial pharyngeal cartilage, altered esophagus morphology, irregular surface epithelium morphology with epithelial blebs, aberrant swim bladder epithelium and impaired fin development [27,31]. Similar to findings in mice, a switch towards expression of mesenchymal IIIc isoforms of *fgfr1a, fgfr1b, fgfr2* and *fgfr3* is detected in double mutant zebrafish embryos [31]. Further, aberrant, non-cartilage cells within the ethmoid plate cleft ectopically express a gene associated with facial clefting across vertebrate species, *irf6* [27]. 

### 3.3. FAM50A

Intellectual developmental disorder, X-linked, syndromic, Armfield type (MRXSA; also known as Armfield X-linked mental retardation syndrome; OMIM 300261) is caused by hemizygous missense mutations in *FAM50A* [32]. The syndrome is characterized by intellectual disability, developmental delay, dysmorphic facial features, skeletal defects (particularly affecting the hands and feet) and ocular anomalies, and commonly includes poor or absent speech, congenital heart defects including atrial septal defect, patent ductus arteriosus, tetralogy of Fallot and right ventricle dilation, brain dysmorphologies, gastrointestinal defects, genitourinary anomalies and neurological defects such as seizures and hypotonia. The dysmorphic facial features can include macrocephaly, prominent, tall and/or broad forehead, bitemporal narrowing, faint hemangiomas between brows and at back of neck, downslanting palpebral fissures, hypotelorism, epicanthal folds, infraorbital creases, proptosis, exotropia, strabismus, keratoconus, nystagmus, depressed nasal bridge, wide nasal root, short and lightly upturned nose with underdeveloped nares, tubular nose, bulbous nose, prominent lips, bow-shaped mouth, cleft palate, hypodontia, microretrognathia, micrognathia, low-set, large and/or slightly posteriorly rotated ears and excessively folded helices [32,33] (Table 1).

In zebrafish, *fam50a* transcripts are ubiquitously expressed throughout gastrulation, but from 1 dpf onwards, expression is restricted to anterior structures including the eyes, mandible and brain. *fam50a*-null embryos exhibit defects in eye, craniofacial cartilage and brain development at 5 dpf and die approximately 1 day later. In particular, *fam50*-null fish display anterior-posterior shortening of the pharyngeal skeleton, delayed branchial arch patterning and a wider ceratohyal angle at 3 dpf compared to controls. These changes are accompanied by increased expression of the p53 pathway effectors *tp53, mdm2* and *cdkn1a* at 2 dpf. The craniofacial cartilage defects of *fam50a*-null fish are recapitulated upon morpholino-mediated knockdown of *fam50a*, with morphants exhibiting increased cell cycle progression and apoptosis in the head region [32].

### 3.4. hnRNP H2

Mental retardation, X-linked, syndromic, Bain type (MRXSB; OMIM 300986) is associated with heterozygous mutation in *HNRNPH2*, the gene encoding the RBP hnRNP H2 [34]. The syndrome is characterized by intellectual disability, poor or absent speech, motor difficulties, growth retardation and musculoskeletal abnormalities, and may also include dysmorphic facial features, epilepsy, neuropsychiatric abnormalities, cortical vision impairment, acquired microcephaly and feeding difficulties [34,35]. The dysmorphic facial features can include short palpebral fissures, almond-shaped eyes, strabismus, long columella, hypoplastic alae nasi, short philtrum, full lower lip and micrognathia [34] (Table 1).

### 3.5. hnRNP K

Au-Kline syndrome (AUKS; OMIM 616580) is caused by heterozygous mutation in *HNRNPK*, the gene encoding the RBP hnRNP K, or a 264 kb deletion in chromosome 9q21.32 encompassing *HNRNPK* [36,37]. The syndrome is characterized by intellectual disability, developmental delay and craniofacial dysmorphologies, and may also include skeletal and connective tissue anomalies, congenital heart malformations, genitourinary anomalies, delayed psychomotor development, speech impairment, cryptorchidism, nuchal skin thickening, hand/foot abnormalities, high pain tolerance, hearing loss, vision abnormalities and agenesis of the corpus callosum [36,37]. The craniofacial dysmorphologies can include dolichocephaly, craniosynostosis, ridged metopic suture, long face, broad or sparse eyebrows, long palpebral fissures, shallow orbits, ptosis, lateral eversion of the lower eyelids, full cheeks, broad nasal bridge, bifid nasal tip, hypoplastic alae nasi, open or down-turned mouth, thin upper lip, high or cleft palate, prominent midline groove of the tongue, missing teeth, malocclusion, low-set, prominent ears, underdeveloped ear helices, preauricular pits and earlobe crease [36,37,38,39] (Table 1).

### 3.6. hnRNP U

Developmental and epileptic encephalopathy 54 (DEE54; OMIM 617391) is caused by heterozygous mutation in *HNRNPU*, the gene encoding the RBP hnRNP U [40]. DEE54 is characterized by delayed psychomotor development, seizures and severe intellectual disability, and may also include hypotonia, absent speech, delayed myelination in the brain, hyperlaxity, enlarged ventricles of the brain, microcephaly, deep-set eyes with epicanthal folds, gray sclerae, narrow palate, short second digit of the hand and autistic features [40,41,42] (Table 1).

### 3.7. Hu Proteins

In mice, *Elavl1* encodes for the RBP HuA (also known as HuR), while *Elavl4* encodes for HuD. Conditional ablation of *Elavl1* in the epiblast lineage using the *Sox2-Cre* driver results in craniofacial and axial skeletal defects, absence of the spleen and lung dysmorphologies. Approximately 80 percent of these embryos exhibit delayed ossification in the neurocranium and upper jaw, and less than 15 percent have nasal clefting [43]. Alternatively, *Elavl4*-null embryos have neuronal development defects, including a transient cranial nerve phenotype characterized by impaired neurite extension of the glossopharyngeal, hypoglossal, trigeminal and acousticofacial nerves at E10.5 [44].

In chick, an anti-panELAV/HU antibody detects expression in the neural tube at the level of the pharyngeal region as early as stage 14. Immunoreactive NC-derived cells are found in the dorsal root ganglia beginning at stage 17 and in the sympathetic ganglia at stage 19 [45].

### 3.8. PTB

In mice, *Ptbp1* encodes the RBP PTB (also known as hnRNP I). By combining null and floxed alleles of *Ptbp1* with the *Nestin-Cre* driver, conditional ablation is achieved in the mouse central and peripheral nervous system. A subset of these *Ptbp1^fl/-^;Nestin-Cre^+/Tg^* postnatal day (P) 21 pups exhibit a domed cranium, in addition to brain abnormalities [46].

In *Xenopus tropicalis*, *ptbp1* is expressed in the ectoderm and mesoderm at the gastrula stage. At neurula stages (stage 16–18), expression is maintained in the ectoderm with strong expression in the neural plate border and NC. Expression is additionally detected in the branchial arches, eyes, otic placodes and pronephros at the tailbud stage (stage 25). Finally, *ptbp1* transcripts in the head region localize to the mandibular, hyoidal and branchial arches, eyes and otic vesicles at the late tailbud stage (stage 34). Expression is additionally detected in migrating trunk NC cells, the ventral aorta, lateral plate and intermediate mesoderm, dermatome and sclerotome at this stage [47].

### 3.9. RBFOX2

In mice, RBFOX2 is expressed in the dorsal neural tube, premigratory and migratory NC cells and somites along the rostrocaudal axis at E9.5. As development proceeds, RBFOX2 becomes restricted to the ventral neural tube. RBFOX2 is additionally detected in the dorsal root ganglia as early as E10.5 and in the trigeminal ganglion and craniofacial mesenchyme at E12.5. Conditional ablation of *Rbfox2* in premigratory and migratory NC cells as well as the somites using the *Pax3-Cre* driver results in cranial nerve deformities at E10.5, subcutaneous edema at E17.5, and clefting of the secondary palate and shortening of the body axis at P1. The cleft palate defect stems from reduced proliferation of the palatal shelf mesenchyme at E12.5–15.5 and a subsequent failure of the palatal shelves to fuse at the midline along the anterior-posterior axis from E15.5-E18.5. The same palatal clefting phenotype is observed upon conditional ablation of *Rbfox2* in the NC lineage using the *Wnt1-Cre2* driver. Both *Rbfox2^fl/fl^;Pax3-Cre^+/Tg^* and *Rbfox2^fl/fl^;Wnt1-Cre2^+/Tg^* embryos at late gestation exhibit hypoplasia and reduced ossification of the majority of NC-derived bones in the frontonasal skeleton and absence of the palatal process of the palatine bone. Additionally, RBFOX2 is able to bind to several transcripts involved in NC or craniofacial development, such as *Map3k7* (encoding MAP3K7/TAK1), *Fn1* and *Postn*, in palatal mesenchyme cells. Conditional ablation of *Rbfox2* using the *Pax3-Cre* driver leads to the differential alternative RNA splicing and subsequent downregulation of expression for a subset of these transcripts in craniofacial tissues. Transcripts encoding TGFβ pathway components are also downregulated in *Rbfox2^fl/fl^;Pax3-Cre^+/Tg^* craniofacial tissues, and phosphorylation of a TGFβ pathway effector, the protein serine/threonine kinase TAK1, and its targets p38 MAPK and SMAD2 are reduced in *Rbfox2^fl/fl^;Pax3-Cre^+/Tg^* palatal shelves. Interestingly, overexpression of TAK1 in cultured palatal mesenchyme cells restores expression of *Map3k7* and *Fn1* and rescues the proliferation defects observed in *Rbfox2^fl/fl^;Pax3-Cre^+/Tg^* embryos. Further, SMAD2 and SMAD3 are able to bind to the *Rbfox2* promoter in response to TGFβ treatment of palatal mesenchyme cells to generate a positive feedback loop [48].

### 3.10. RBM10

TARP syndrome (TARPS; OMIM 311900) is caused by hemizygous mutation in *RBM10* [49]. The syndrome is variably characterized by talipes equinovarus, atrial septal defect, Robin sequence (cleft palate, glossoptosis and micrognathia) and persistent left superior vena cava [50]. Additional fully-penetrant features include cognitive, motor and language delays, brain malformations, neurological defects, pulmonary abnormalities, failure to thrive and facial dysmorphism. Patients commonly exhibit cardiac and distal limb anomalies beyond those described in the TARP acronym, postnatal growth delay, skeletal defects and gastrointestinal abnormalities, with a subset of patients additionally displaying genital malformations, renal abnormalities, vision impairment and hearing loss [51,52]. The dysmorphic facial features can include large fontanels, round face, hypertelorism, underdevelopment of the alae nasi, prominent columella, wide mouth, downturned mouth corners, high and/or narrow palate and low-set and/or posteriorly rotated ears [52] (Table 1).

Consistent with its role in human development, *Rbm10* transcripts are enriched in the first and second pharyngeal arches, limb buds and tailbud of E9.5-E10.5 mouse embryos. Expression of *Rbm10* transcripts in the pharyngeal arches decreases by E11.5 [49]. *Rbm10* knockout mouse embryonic stem cells as well as mouse mandibular cells isolated from E11.5 embryos exhibit decreased growth compared to wild-type cells [53]. 

### 3.11. RBMX

Mental retardation, X-linked, syndromic 11 (MRXS11; also known as Shashi X-linked mental retardation syndrome; OMIM 300238) is associated with a hemizygous mutation in the gene encoding the RBP RBMX (also known as hnRNP G) [54]. The syndrome is characterized by moderate mental retardation, craniofacial dysmorphologies, obesity and enlarged testes. The craniofacial dysmorphologies include coarse facies, prominent supraorbital ridges, narrow palpebral fissures, puffy eyelids, bulbous nose, prominent lower lip and large ears [55] (Table 1).

In *Xenopus laevis*, *rbmx* transcripts are expressed in the neural tube, NC and somites at stages 20–25. Morpholino-mediated knockdown of *rbmx* results in decreased expression of the NC marker *slug* at stages 17 and 35, as well as eye defects, jaw deformities, loss of the trigeminal ganglion and a reduction in melanocyte numbers in the trunk at tadpole stages, among other neural and axial muscle segmentation defects [56]. 

In zebrafish, *rbmx* transcripts are ubiquitously expressed during gastrulation, but become restricted to multiple sites as development progresses, including the branchial arches from 2–7 dpf. Similar to the findings in *Xenopus laevis*, morpholino-mediated knockdown of *rbmx* in zebrafish results in a reduction in the size of the head and eyes, mild cyclopia and absent jaws at 2 dpf, in addition to defects in brain development and somite patterning [57].

### 3.12. SRSF3

In mice, *Srsf3* is broadly expressed during development, with transcripts enriched in the head region and facial processes from E8.5-E10.5. At these same timepoints, SRSF3 protein co-localizes with SOX10 in NC cells migrating away from the cranial neural folds and is expressed in the mesenchyme and overlying ectoderm of the pharyngeal arches and facial processes. Homozygous, conditional ablation of *Srsf3* in the murine NC lineage using the *Wnt1-Cre* driver results in hypoplastic facial processes, increased distance between the nasal pits and brain malformations at E10.5. At E12.5, these conditional knockout embryos additionally exhibit facial clefting and hypoplasia of the secondary palatal shelves and tongue. A subset of embryos at these stages displays facial subepidermal blebbing, facial hemorrhaging and a wavy neural tube. By E18.5, *Srsf3* conditional knockout embryos have a reduced head size, with anterior bones and cartilages that are clefted and hypoplastic and midface elements that are more severely affected and, in some cases, missing entirely. The mandible, coronoid process and Meckel’s cartilage are hypoplastic, the hyoid is not ossified or severely hypoplastic, the lesser and greater horns of the hyoid as well as the thyroid and cricoid cartilages are hypoplastic and the tracheal cartilage rings are misshapen. These defects stem from reduced proliferation of the mesenchyme underlying the cranial neural folds at E8.0 and increased cell death in the mesenchyme of pharyngeal arch 1 at E9.5. Additionally, conditional ablation of *Srsf3* in the murine NC lineage leads to a significant increase in expression of multiple genes involved in NC differentiation, including *Col2a1* and *Zic5* [58]. PI3K is the primary effector of PDGFRα signaling during skeletal development in the mouse, and disruption of this axis leads to palatal clefting [59,60]. SRSF3 is phosphorylated at AKT consensus sites in response to PI3K-mediated PDGFRα signaling in immortalized mouse embryonic palatal mesenchyme cells, driving translocation of phosphorylated SRSF3 into the nucleus where alternative RNA splicing occurs. Relatedly, SRSF3 coordinates the alternative RNA splicing of transcripts encoding protein serine/threonine kinases in the E11.5 maxillary process mesenchyme to regulate PDGFRα-dependent intracellular signaling, specifically, phosphorylation of ERK1/2 [58]. 

In *Xenopus laevis*, *srsf3* is expressed ubiquitously, with transcripts enriched in the neural tube and somites at stages 20–32. Further, overexpression of *srsf3* mRNA leads to a change in the shape and location of the expression domain of the NC marker *slug*, among other defects in neurogenesis and mesoderm development [56].

## 4. RBPs Involved in Trafficking 

### VICKZ Proteins

In mice, *Igf2bp1* encodes for the RBP IMP-1 (also known as VICKZ1 and ZBP1). At E10.5, *Igf2bp1* transcripts are expressed in the forebrain, hindbrain, medial and lateral nasal processes, first and second pharyngeal arches, limb buds and tail. By E12.5, expression expands into the neural tract, eyes, tongue, heart, lung, liver and somites. At E17.5, *Igf2bp1* transcripts primarily localize to the intestine, kidney and liver. *Igf2bp1*-null pups have a reduced body size, a short snout and, in a subset of mice, a kinked tail, wide-spread cartilage loss and dysmorphologies of the intestine, kidney and liver. Within the craniofacial region, cartilage loss was detected in the nasal bone and mandible [61]. Of note, in addition to its role in trafficking, IMP-1 is also involved in translation.

While the chick VICKZ paralogs are expressed in the neural epithelium and neural folds at the 5 somite stage, their expression is reduced in delaminating and migrating cranial NC cells at the 8 and 13 somite stages, respectively. Morpholino-mediated knockdown or expression of dominant-negative VICKZ at the 2–4 somite stage leads to enhanced cranial NC cell delamination, while this effect is reversed upon overexpression of VICKZ. Further, down-regulation of VICKZ activity decreases expression of Integrin α6 in the neural tube, while overexpression of VICKZ results in abnormal persistence of Integrin α6 expression at this site. Relatedly, immunoprecipitation of VICKZ from 3–4 day embryos leads to an enrichment in *ITGA6* mRNA, suggesting that the two molecules are part of the same complex in chick embryos [62].

## 5. RBPs Involved in Translation 

### 5.1. DDX3X

Intellectual developmental disorder, X-linked, syndromic, Snijders Blok type (MRXSSB; OMIM 300958) is caused by heterozygous mutations in *DDX3X* in females or hemizygous missense mutations in *DDX3X* in males [63]. In female patients, the syndrome is characterized by intellectual disability or developmental delay and can also include neurological abnormalities such as hypotonia, behavior problems, movement disorder and epilepsy, as well as microcephaly, dolichocephaly, brain dysmorphologies, joint hyperlaxity, skin pigment abnormalities, visual impairment, precocious puberty, scoliosis, hearing loss, cleft lip and/or palate and neuroblastoma. A subset of patients exhibits dysmorphic facial features, which commonly include long and/or hypotonic face, high and/or broad forehead, hypertelorism, wide nasal bridge, bulbous nose, narrow alae nasi and anteverted nostrils [63,64,65] (Table 1). Less common dysmorphic facial features include telecanthus, short palpebral fissures, epicanthal folds, smooth and/or long philtrum, thin upper vermilion, high-arched palate, micrognathia and large, protruding or low-set ears [64,66]. In male patients, the syndrome is characterized by intellectual disability and can also include neurological abnormalities such as behavior problems and movement disorder, brachycephaly, macrocephaly, facial dysmorphisms, strabismus, bifid uvula, nuchal thickening, pulmonary stenosis, cardiac anomalies, visual impairment and hearing loss [63,67,68] (Table 1). 

In mice, though DDX3X protein is highly expressed throughout the embryo at E7.5, its expression becomes restricted as development proceeds, such that by E14.5, protein expression localizes to the brain, trigeminal ganglion, eyes, nasal cavity, lung, heart and liver. Conditional ablation of *Ddx3x* in the mouse epiblast using the *Sox2-Cre* driver results in male embryos exhibiting developmental delay and abnormal head fold morphology at E9.5, and neural tube closure defects, an underdeveloped brain and reduced head size at E11.5, among other allantois and myocardial trabeculae defects. These changes are accompanied by embryo-wide increases in mitosis at E9.5 and apoptosis at E8.5-E9.5. Moreover, *Ddx3x^fl^/Y;Sox2-Cre^+/Tg^* embryos exhibit increased DNA damage, p53 phosphorylation and expression of transcripts encoding proteins involved in cell cycle progression at E9.5 [69].

In *Xenopus tropicalis*, *ddx3* is ubiquitously expressed at early gastrula stages in the animal half of embryos. By the end of gastrulation, expression is predominantly localized to the dorsal ectoderm. Expression of *ddx3* is further restricted to the neural plate and neural plate border during neurulation, and to the head and somites at tailbud stages. Morpholino-mediated knockdown of *ddx3* results in hypoplasia of the craniofacial cartilage at approximately stage 46. These defects stem from reduced expression of transcripts encoding proteins involved in neural plate border specification, including *pax3* and *msx1*, and NC specification, including *snai2* and *sox9*, at the end of gastrulation. Moreover, the function of Ddx3 in this context is mediated by the downstream effector Rac1, leading to activation of Akt, the subsequent phosphorylation and inhibition of Gsk3β, and ultimately the stabilization of two proteins required for NC induction, β-catenin and Snai1 [70]. 

Similarly, morpholino-mediated knockdown of zebrafish *ddx3xb* results in reduced head and brain sizes at 2 dpf [71]. DDX3X has been shown to regulate WNT/β-catenin signaling [72] and co-injection of *wnt3a* and *ddx3xb* mRNA into zebrafish embryos results in a more severe ventralization phenotype at 2 dpf than that observed upon injection of *wnt3a* alone. This effect is maintained upon injection of *ddx3xb* mRNA harboring mutations found in human male, but not female, MRXSSB patients, indicating that the *de novo* variants in the latter case are loss-of-function mutations with respect to regulation of the Wnt pathway [63]. 

### 5.2. FMRP

The majority of Fragile X syndrome (FXS; OMIM 300624) cases are caused by a full trinucleotide (CGG) repeat expansion of more than 200 repeats in the 5′ untranslated region of *FMR1* that results in hypermethylation of *FMR1* and loss of expression of the RBP FMRP [73,74,75,76]. Human *FMR1* transcripts are expressed in the neural tube, central nervous system and several non-neural tissues, among them the eyes, branchial arches and cartilages, at 3–7 weeks of gestation [77]. Accordingly, FXS is characterized by mental retardation, macroorchidism and dysmorphic facial features, which can include long, narrow face, high and/or broad forehead, broad palpebral fissures, midface hypoplasia, flat nasal bridge, broad nose, upward displacement of the nasal alae, broad philtrum, large mouth, thick lips, long upper middle incisors, high-arched palate, prognathia, hypotonia affecting the lower jaw and large, low-set and/or poorly formed ears [78,79,80,81,82] (Table 1). 

In mice, *Fmr1* transcripts are ubiquitously expressed at E10.5, with strong expression in the neural tube. By adulthood, however, high expression is restricted to the brain, eyes, testes, ovaries, thymus, esophagus and spleen [83]. Similar to the phenotypes described for human FXS patients, *Fmr1*-null mice exhibit learning deficits, hyperactivity, macroorchidism and craniofacial defects including reduced skull vault height, decreased length and rear width of the inner skull, interparietal bone shortening and reduced mandible width [82,84]. Further, the NC regulator AP-2α binds the *FMR1* promoter in human HeLa cells and *Fmr1* expression is reduced in E18.5 head tissue of *Tfap2a*-null embryos [85].

In *Xenopus laevis*, *fmr1* transcripts are expressed in the neural tube, migrating cranial NC cells and eye anlagen at stage 20. Transcripts localize to the central nervous system, NC, eyes and pharyngeal arches at early tailbud stages 25–26. *fmr1* expression increases in the central nervous system and within the craniofacial region, including the eyes and otic vesicle, by the late tailbud stage, with additional expression in the notochord [86,87]. Morpholino-mediated knockdown of *fmr1* in an animal-dorsal blastomere at the 8-cell stage results in smaller or absent eyes and reduced and dysmorphic cranial cartilage elements. The latter phenotypes are associated with defects in NC migration as indicated by decreased expression of *foxd3* at stage 20 as well as *krox20* and *twist* at stage 23 [87]. Consistent with findings in mice, *fmr1* and *tfap2a* transcripts co-localize in the pharyngeal arches and craniofacial regions of *Xenopus leavis* embryos at stage 25. Moreover, overexpression of a dominant-negative Ap-2α and inhibition of Ap-2α through Chordin overexpression lead to significant repression of *fmr1* expression in animal cap assays [85].

Morpholino-mediated knockdown of *fmr1* in zebrafish leads to abnormal axon branching, reduced trigeminal neuron number and dysmorphic craniofacial cartilage. Specifically, Meckel’s cartilage is shorter and wider at 5 dpf in morphants and the angles between the ceratobranchial/ceratohyal arch cartilage and basibranchial/hyal cartilage and between Meckel’s cartilage and the rostrocaudal axis are less acute in morphants. Further, expression of a marker of migrating cranial NC cells, *dlx-2a*, is reduced and mis-localized in the first and second pharyngeal arches of morphants at 22 hpf [88]. Importantly, however, the phenotypes observed in morphants are not recapitulated in *fmr1*-null zebrafish, which are viable and fertile with no apparent defects nor altered expression of *dlx-2a* [89].

### 5.3. PACT

Dystonia 16 (DYT16; OMIM 612067) is caused by homozygous mutations in *PRKRA*, the gene encoding the RBP PACT in humans [90]. DYT16 is characterized by early-onset, progressive dystonia and gait abnormalities, and may also include delayed development, oromandibular dystonia, laterocollis/retrocollis, spasmodic dysphonia, bradykinesia, dysphagia, opisthotonic posturing, facial grimacing, speech and language impairment, leg pain, blepharospasm and hypomimia [90,91] (Table 1).

In mice, *Prkra* transcripts are expressed in mouse blastocysts, the developing ears as early as E12 and in the anterior skull base and mandible at E16 [92,93]. Consistently, *Prkra*-null embryos die before implantation [93]. However, mice in which a neomycin resistance cassette replaces the sequence in exon 8 encoding the PKR activation domain express no PRKRA protein (also called RAX in mice) and exhibit hypoplasia of the anterior pituitary lobe, reduced postnatal body weight, fertility defects and a number of craniofacial defects, including wide cranial sutures, hypoplastic interparietal bone, prominent bossing of the forehead, shortened nose, hypolastic nasal turbinates, hypoplastic mandibular condyle and microtia [92,94,95]. The pituitary defects stem from reduced proliferation in the anterior pituitary lobe as early as P21 [94]. The ear defects are characterized by hypoplastic pinnae and external auditory canals, malformed middle ear ossicles and reduced middle ear space and tympanic membrane, resulting in hearing impairment [92]. 

## 6. Conclusions

Multiple screens have revealed a prevalence of transcripts encoding RBPs in mouse craniofacial tissues. One such study identified an enrichment of transcripts encoding 17 proteins with RNA recognition motifs in mouse E10.5 pharyngeal arches 1 and 2 [96]. A more recent study revealed that transcripts encoding 497 RBPs are expressed in mouse craniofacial process ectoderm and/or mesenchyme from E10.5-E12.5, including *Esrp1*, *Esrp2*, *Elavl4*, *Rbfox2*, *Rbm10*, *Rbmx*, *Igf2bp1* and *Fmr1* [97], each of which is discussed above. However, in contrast to the well-studied roles of signaling molecules and transcription factors in regulating NC and craniofacial development, only a handful of the 1,393 RBPs identified in humans [2] and the 1,914 RBPs identified in mouse [2] have been examined in these contexts. The study of RBPs thus offers a relatively untapped opportunity to provide significant insight into the mechanisms underlying gene expression regulation during craniofacial morphogenesis.

As the majority of this review focuses on splicing regulators not associated with the spliceosome, it is useful to compare the phenotypes observed upon mis-expression of core components of the spliceosome and these additional splicing regulators. Spliceosomopathies are characterized by microcephaly, malar hypoplasia, eye anomalies, micrognathia and external ear anomalies within the craniofacial region, as well as intellectual disability, psychomotor delay, heart defects and limb abnormalities [5]. As discussed above, with the exception of malar hypoplasia, each of these phenotypes is observed upon mutation of one or more non-spliceosome-associated RBP. Given the prevalence of craniofacial dysmorphologies observed in spliceosomopathies, it has been hypothesized that NC cells have an increased need for alternative RNA splicing, perhaps owing to their multipotency, and that mutation of core components of the spliceosome leads to a common mechanism of p53 pathway disruption and NC cell apoptosis [5]. Interestingly, with the exception of EIF4A3 [98], modulation of the p53 pathway has not been reported for any of the splicing regulators discussed here, indicating that these RBPs may affect cell survival through alternative pathways.

While the target transcripts of individual RBPs do not appear to overlap in the context of craniofacial morphogenesis, even within related tissues, common cellular mechanisms are regulated by these RBPs to contribute to proper development. For example, only six transcripts are commonly differentially alternatively spliced (*Exoc1*, *Meg3*, *Nfya*, *Smarca2*, *Tpm1* and *Uap1*) and only two transcripts are commonly differentially expressed (*Lama5* and *Thrb*) between E12.5 *Rbfox2^fl/fl^;Pax3-Cre^+/Tg^* craniofacial tissue and E11.5 *Srsf3^fl/fl^;Wnt1-Cre^+/Tg^* maxillary process mesenchyme tissue relative to their respective controls [48,58]. Notably, however, knockdown or ablation of *Esrp1/2*, *Rbfox2*, *Srsf3* and Ddx3x each results in altered expression of transcripts encoding components of developmental signaling pathways. These include the FGF, WNT, SHH, BMP and TGFβ signaling pathways in the case of ESRP1/2 [30], and the TGFβ, PDGF and Wnt signaling pathways for RBFOX2 [48], SRSF3 [58] and Ddx3x [70,72], respectively. These findings raise the intriguing possibility that RNA processing mediated by RBPs serves to regulate both intracellular and intercellular signaling during craniofacial development, often in feedback loops involving developmental signaling pathways and the phosphorylation of intracellular effectors. Further, mis-expression of several of the RBPs discussed here results in reduced cell proliferation (in the cases of CNBP, ESRP1, RBFOX2, RBM10, SRSF3 and PACT/RAX) and/or increased cell death (in the cases of Cnbp, Eif4a3, Fam50a, SRSF3 and DDX3X) across vertebrate species, indicating that these are common mechanisms underlying the observed craniofacial phenotypes. 

Several outstanding questions remain regarding RBPs in NC and craniofacial development: Where and when are RBPs expressed and what upstream factors regulate this expression? How is the activity of RBPs regulated by post-translational modifications? Which proteins interact with RBPs to enhance or antagonize function, and to what extent do RBPs cross-regulate one another? Similarly, how does the network of RBPs expressed in a given cell type establish and/or reinforce a tissue-specific transcriptome? Which are the direct target transcripts of RBPs and how does mis-expression of these transcripts contribute to a disease phenotype? As highlighted above, animal models of individual RBPs provide a framework to address many of these questions. Undoubtedly, the advent of new genetic, molecular and biochemical approaches promises to reveal critical roles for additional RBPs both during development and disease.

## Figures and Tables

**Figure 1 jdb-09-00034-f001:**
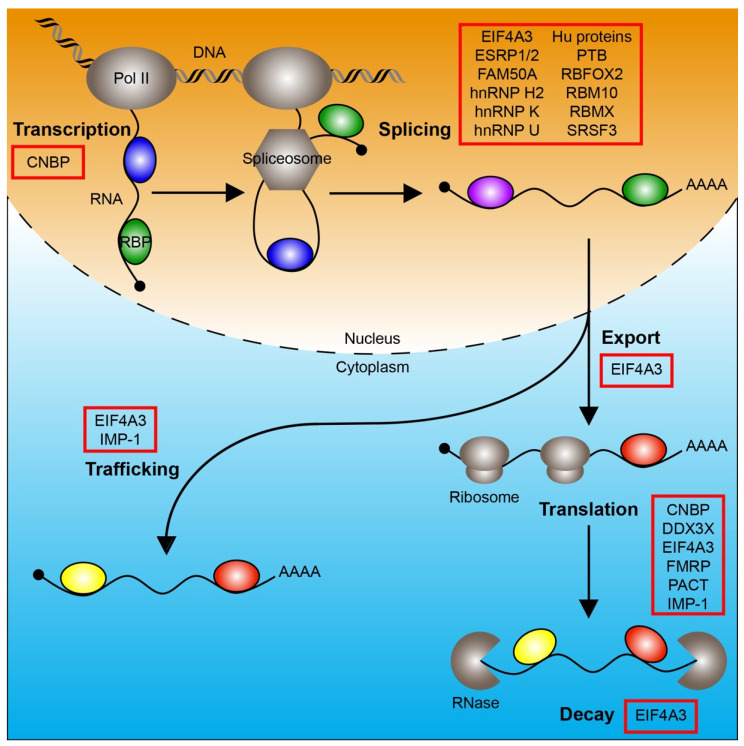
RNA-binding proteins regulate RNA metabolism. The processes of transcription, splicing, nuclear export, trafficking, translation and decay are depicted. RNA-binding proteins (colored ovals) that regulate these processes and have a role in mammalian neural crest and/or craniofacial development are boxed in red. Pol II, RNA polymerase II; RBP, RNA-binding protein.

**Table 1 jdb-09-00034-t001:** Craniofacial defects associated with mis-expression of human RBPs.

Disorder (Associated RBP)	Craniofacial Defect(s)
DM2 (CNBP)	Hearing loss
MRXSSB (DDX3X)	Females: microcephaly; dolichocephaly; long and/or hypotonic face; high and/or broad forehead; hypertelorism; visual impairment; wide nasal bridge; bulbous nose; narrow alae nasi; anteverted nostrils; cleft lip and/or palate; hearing loss Males: brachycephaly; macrocephaly; strabismus; bifid uvula; visual impairment; hearing loss
Robin sequence with cleft mandible and limb anomalies (EIF4A3)	Microstomia; cleft palate; absent lower central incisors; micrognathia; abnormal fusion of the mandible; ear anomalies
DFNB109 (ESRP1)	Hearing loss; vestibular dysplasia
Non-syndromic cleft lip with or without cleft palate (ESRP2)	Cleft lip and/or palate
MRXSA (FAM50A)	Macrocephaly; prominent, tall and/or broad forehead; bitemporal narrowing; faint hemangiomas between brows; downslanting palpebral fissures; hypotelorism; epicanthal folds; infraorbital creases; proptosis; exotropia; strabismus; keratoconus; nystagmus; depressed nasal bridge; wide nasal root; short and lightly upturned nose with underdeveloped nares; tubular nose; bulbous nose; prominent lips; bow-shaped mouth; cleft palate; hypodontia; microretrognathia; micrognathia; low-set, large and/or slightly posteriorly rotated ears; excessively folded helices
FXS (FMRP)	Long, narrow face; high and/or broad forehead; broad palpebral fissures; midface hypoplasia; flat nasal bridge; broad nose; upward displacement of the nasal alae; broad philtrum; large mouth; thick lips; long upper middle incisors; high-arched palate; prognathia; hypotonia affecting the lower jaw; large, low-set and/or poorly formed ears
MRXSB (hnRNP H2)	Microcephaly; short palpebral fissures; almond-shaped eyes; strabismus; long columella; hypoplastic alae nasi; short philtrum; full lower lip; micrognathia
AUKS (hnRNP K)	Dolichocephaly; craniosynostosis; ridged metopic suture; long face; broad or sparse eyebrows; long palpebral fissures; shallow orbits; ptosis; lateral eversion of the lower eyelids; vision abnormalities; full cheeks; broad nasal bridge; bifid nasal tip; hypoplastic alae nasi; open or down-turned mouth; thin upper lip; high or cleft palate; prominent midline groove of the tongue; missing teeth; malocclusion; low-set, prominent ears; underdeveloped ear helices; preauricular pits; earlobe crease; hearing loss
DEE54 (hnRNP U)	Microcephaly; deep-set eyes with epicanthal folds; gray sclerae; narrow palate
DYT16 (PACT)	Oromandibular dystonia; facial grimacing; blepharospasm; hypomimia
TARPS (RBM10)	Large fontanels; round face; hypertelorism; vision impairment; underdevelopment of the alae nasi; prominent columella; wide mouth; downturned mouth corners; high, narrow and/or cleft palate; glossoptosis; micrognathia; low-set and/or posteriorly rotated ears; hearing loss
MRXS11 (RBMX)	Coarse facies; prominent supraorbital ridges; narrow palpebral fissures; puffy eyelids; bulbous nose; prominent lower lip; large ears

## Data Availability

Not applicable.

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
