# Peer review of "The Role of RNA-Binding Proteins in Vertebrate Neural Crest and Craniofacial Development"

_jdb, 2021, doi:10.3390/jdb9030034_

Round 1

Reviewer 1 Report

This review is overall very well written and describes in detail different RNA-binding proteins in craniofacial development. I have a few suggestions to improve the manuscript:

  • The title is ‘The role of RNA-binding proteins in mammalian neural crest and craniofacial development’, yet, in each section, the authors systematically review data on RBPs in non-mammalian vertebrates. Therefore, I would suggest a few different options: 1) Focus only on mammalian models and keep that title; 2) keep the current content of the review but change the title and make it more general to vertebrates; 3) keep the major focus of the review on mammalian models and combine all the data on non-mammalian vertebrate models in one final section of the review that will explore the evolutionary conservation and/or divergence of these proteins in vertebrates.
  • Although the use of English is excellent, on many occasions throughout the manuscript, the authors tend to use overly long sentences. This makes it difficult for the reader to digest the content of the review. An example of such long sentence is in section 2.1, lines 110-113 (Knockdown of […] tail curvature). Breaking down long sentences into two or more, and using more linking words, would significantly improve in the readability and digestibility of this manuscript.
  • In this manuscript, each section is composed of a descriptive list of gene expression and mutant phenotypes for one RBP in different models. Summarizing main findings for each section with a small paragraph would help in the digestibility of the data presented in this review.

Author Response

Reviewer #1

This review is overall very well written and describes in detail different RNA-binding proteins in craniofacial development.

We thank the Reviewer for these supportive comments.

  1. The title is ‘The role of RNA-binding proteins in mammalian neural crest and craniofacial development’, yet, in each section, the authors systematically review data on RBPs in non-mammalian vertebrates. Therefore, I would suggest a few different options: 1) Focus only on mammalian models and keep that title; 2) keep the current content of the review but change the title and make it more general to vertebrates; 3) keep the major focus of the review on mammalian models and combine all the data on non-mammalian vertebrate models in one final section of the review that will explore the evolutionary conservation and/or divergence of these proteins in vertebrates.

We have changed the title to “The role of RNA-binding proteins in vertebrate neural crest and craniofacial development” at the Reviewer’s suggestion.

  1. Although the use of English is excellent, on many occasions throughout the manuscript, the authors tend to use overly long sentences. This makes it difficult for the reader to digest the content of the review. An example of such long sentence is in section 2.1, lines 110-113 (Knockdown of […] tail curvature). Breaking down long sentences into two or more, and using more linking words, would significantly improve in the readability and digestibility of this manuscript.

We have made every attempt to limit our use of long sentences to instances in which we discuss a list of phenotypes, as in the sentence highlighted by the Reviewer. We believe that breaking up these sentences would decrease readability, as there is often not a natural segregation of phenotypes within these lists that would warrant breaking them up into two sentences. However, wherever possible, we have listed “craniofacial defects” among the phenotypes in the first sentence and then provided more detail on the craniofacial defects in a second, subsequent sentence.

  1. In this manuscript, each section is composed of a descriptive list of gene expression and mutant phenotypes for one RBP in different models. Summarizing main findings for each section with a small paragraph would help in the digestibility of the data presented in this review.

Among the 17 proteins discussed in the review, the majority (11 proteins) are discussed in only one or two species. Wherever possible, we have drawn connections between RBP expression patterns and mutant phenotypes among species. Further, we have drawn connections between several of the discussed RBPs in the Conclusions section. At the suggestion of Reviewer #3, we have grouped the text to match the outline provided in Figure 1. As such, proteins have been grouped as being involved in transcription, splicing, trafficking and translation. We feel that this change has substantially enhanced the readability of the review.

Reviewer 2 Report

The topic covered by this review is of broad interest to the field of developmental biology, particularly neural crest and craniofacial development. It is well-written, comprehensive and well-illustrated.

Minor:

  • First sentence of introduction: change “mammalian” to “vertebrate”
  • For zebrafish homologues it would be helpful to know which alleles are under consideration in each section, considering that teleosts have duplicated genomes.
  • I was surprised to see chicken gene nomenclature in all caps – isn’t this reserved for human genes?

Author Response

Reviewer #2

The topic covered by this review is of broad interest to the field of developmental biology, particularly neural crest and craniofacial development. It is well-written, comprehensive and well-illustrated.

We thank the Reviewer for these supportive comments.

  1. First sentence of introduction: change “mammalian” to “vertebrate”.

We have edited the first sentence to read “During vertebrate embryogenesis, neural crest (NC) cells are induced at the neural plate border, undergo an epithelial to mesenchymal transition to delaminate from the neural folds or neural tube, migrate in stereotypical streams throughout the embryo and eventually differentiate into a wide array of derivatives” at the Reviewer’s suggestion.

  1. For zebrafish homologues it would be helpful to know which alleles are under consideration in each section, considering that teleosts have duplicated genomes.

We thank the Reviewer for this suggestion. We have now changed all mentions of “Cnbp” in zebrafish to “Cnbpa” and all mentions of “Ddx3x” to “Ddx3xb”. We checked ZFIN (http://zfin.org) and the following RNA-binding proteins are encoded by a single gene: Eif4a3, Esrp1, Esrp2, Fam50a, Rbmx and Fmr1.

  1. I was surprised to see chicken gene nomenclature in all caps – isn’t this reserved for human genes?

For chick, all gene, RNA and protein names are written according to the Chicken Gene Nomenclature Consortium ( http://birdgenenames.org/cgnc/guidelines). This convention is also observed on UniProt (http://www.uniprot.org).

Reviewer 3 Report

RNA binding proteins (RBPs) are a critical group of biomolecules that regulate the fate of cellular RNAs in multiple stages including transcription, splicing, transport, localization and stability. Regulation of RNAs by these RBPs are important to maintain the proper function and development of animals. The review paper titled “The role of RNA-binding proteins in mammalian neural crest and craniofacial development” is a very timely, comprehensive, and well-written review of the current knowledge on some important RBPs in mammalian craniofacial morphogenesis. I commend the authors for taking such a deep dive on disease relevant to these RBP biology and discuss the knowledge on other models including chicken, Xenopus, and zebrafish in addition to mammals. I think it is a great resource that will be well appreciated by the RNA biology and development research community. I have a few minor comments below but otherwise think the authors have done a great job to cover so much of the literature.

Here are my minor comments to authors.

  1. The authors briefly touch about how these RBPs bind to RNAs using canonical RNA binding domains (RBDs) in the Introduction section. Adding information about the RBDs of each of the RBPs discussed in this paper, will further improve the value of this review article.
  2. Figure 1 is clearly designed (grouped based on function) to give the reader to an overall idea about the RBPs that are going to be discussed in the paper. However, the text discussing the RBPs didn’t follow the outline provided in Figure 1. Authors can move all the proteins that fall into the same category together to ease the reading instead of reporting the proteins in the alphabetical order. For example, all the RBPs within the splicing category can be discussed together and can be wrapped up with a few sentences about splicing and these RBPs.
  3. Can the authors make a schematic representation of all the discussed proteins to enhance the knowledge of the reader? The authors can select one species to make the schematic.

Author Response

Reviewer #3

RNA binding proteins (RBPs) are a critical group of biomolecules that regulate the fate of cellular RNAs in multiple stages including transcription, splicing, transport, localization and stability. Regulation of RNAs by these RBPs are important to maintain the proper function and development of animals. The review paper titled “The role of RNA-binding proteins in mammalian neural crest and craniofacial development” is a very timely, comprehensive, and well-written review of the current knowledge on some important RBPs in mammalian craniofacial morphogenesis. I commend the authors for taking such a deep dive on disease relevant to these RBP biology and discuss the knowledge on other models including chicken, Xenopus, and zebrafish in addition to mammals. I think it is a great resource that will be well appreciated by the RNA biology and development research community. I have a few minor comments below but otherwise think the authors have done a great job to cover so much of the literature.

We thank the Reviewer for these supportive comments.

  1. The authors briefly touch about how these RBPs bind to RNAs using canonical RNA binding domains (RBDs) in the Introduction section. Adding information about the RBDs of each of the RBPs discussed in this paper, will further improve the value of this review article.

We thank the Reviewer for this suggestion. We have added the following sentence to the Introduction: “These proteins contain one or more RBDs, including: RNA recognition motif (ESRP1/2, hnRNP H2, Hu proteins, PTB, RBFOX2, RBM10, RBMX, SRSF3 and IMP-1), zinc-finger domain (CNBP, hnRNP H2 and RBM10), K homology domain (hnRNP K, IMP-1 and FMRP), DEAD motif (EIF4A3 and DDX3X), ROKNT domain (hnRNP K), SAP domain (hnRNP U), Fox-1_C domain (RBFOX2), G-patch domain (RBM10), RBM1CTR domain (RBMX), Agenet domain (FMRP), FXR1P_C domain (FMRP) and double-stranded RNA-binding motif (PACT) [4].”

  1. Figure 1 is clearly designed (grouped based on function) to give the reader to an overall idea about the RBPs that are going to be discussed in the paper. However, the text discussing the RBPs didn’t follow the outline provided in Figure 1. Authors can move all the proteins that fall into the same category together to ease the reading instead of reporting the proteins in the alphabetical order. For example, all the RBPs within the splicing category can be discussed together and can be wrapped up with a few sentences about splicing and these RBPs.

We thank the Reviewer for this suggestion. We have grouped the text to match the outline provided in Figure 1. As such, proteins have been grouped as being involved in transcription, splicing, trafficking and translation. At the first mention of CNBP, EIF4A3 and IMP-1, which are involved in multiple processes, we have highlighted the other processes that each protein contributes to. For example, “Of note, in addition to its role in transcription, CNBP is also involved in translation.”

  1. Can the authors make a schematic representation of all the discussed proteins to enhance the knowledge of the reader? The authors can select one species to make the schematic.

We feel that Figure 1, which depicts the processes of RNA metabolism that each RNA-binding protein (RBP) contributes to, and Table 1, which discusses the craniofacial defects associated with mis-expression of human RBPs, clearly illustrates many of the major points discussed in the review. Given that even within a single species the expression of the discussed RBPs spans several locations and timepoints, and that the phenotypes observed upon mutation of these RBPs leads to defects in multiple tissue types at various timepoints through misregulation of a wide array of cellular activities, summarizing all of this information in a single figure is an almost impossible task.